# Effect of the Financial Crisis on Socioeconomic Inequalities in Mortality in Small Areas in Seven Spanish Cities

**DOI:** 10.3390/ijerph17030958

**Published:** 2020-02-04

**Authors:** Mercè Gotsens, Josep Ferrando, Marc Marí-Dell’Olmo, Laia Palència, Xavier Bartoll, Ana Gandarillas, Pablo Sanchez-Villegas, Santi Esnaola, Antonio Daponte, Carme Borrell

**Affiliations:** 1Agència de Salut Pública de Barcelona, 08023 Barcelona, Spain; 2Institut d’Investigació Biomèdica (IIB Sant Pau), 08041 Barcelona, Spain; 3CIBER Epidemiología y Salud Pública (CIBERESP), 28029 Madrid, Spain; 4Dirección General de Salud Pública, Consejería de Sanidad, Comunidad de Madrid, 28035 Madrid, Spain; 5Observatorio de Salud y Medio Ambiente de Andalucía, Escuela Andaluza de Salud Pública, 18080 Granada, Spain; 6Department of Health of the Basque Country, 01006 Vitoria-Gasteiz, Spain; 7Department of Experimental and Health Sciences, Universitat Pompeu Fabra, 08002 Barcelona, Spain

**Keywords:** socioeconomic factors, trends, mortality, inequalities, urban areas, small areas

## Abstract

Background: The aim of this study was to analyze the trend in socioeconomic inequalities in mortality in small areas due to several specific causes before (2001–2004, 2005–2008) and during (2009–2012) the economic crisis in seven Spanish cities. Methods: This ecological study of trends, with census tracts as the areas of analysis, was based on three periods. Several causes of death were studied. A socioeconomic deprivation index was calculated for each census tract. For each small area, we estimated standardized mortality ratios, and controlled for their variability using Bayesian models (sSMR). We also estimated the relative risk of mortality according to deprivation in the different cities, periods, and sexes. Results: In general, a similar geographical pattern was found for the socioeconomic deprivation index and sSMR. For men, there was an association in all cities between the deprivation index and all-cause mortality that remained stable over the three periods. For women, there was an association in Barcelona, Granada, and Sevilla between the deprivation index and all-cause mortality in the third period. Patterns by causes of death were more heterogeneous. Conclusions: After the start of the financial crisis, socioeconomic inequalities in total mortality in small areas of Spanish cities remained stable in most cities, although several causes of death showed a different pattern.

## 1. Introduction

The impact of economic crises on populations’ health depends on various factors, such as the institutional, cultural, and socioeconomic context of the society, the severity and duration of the crisis, and the responses articulated to combat it [1,2,3,4,5,6,7]. Crises may be accompanied by a higher prevalence of poor health or premature mortality, particularly in vulnerable population subgroups [8,9]. However, some research has described how recessions might improve health, at least in the short term, with a pro-cyclical worsening of mortality during periods of growth and improvement during recessions, mainly due to fewer traffic injuries and alcohol-related deaths [10]. Catalano et al. (2011) [11] described the mechanisms related to changes in health outcomes during economic recessions, most of which are due to job loss and financial strain, as well as stress, frustration, and aggression, and the investment of time and effort in managing the sequelae of job or income loss.

In the European Union, the financial crisis started in 2008. During this period, no change was observed in either all-cause mortality or its decreasing tendency [12,13,14]. However, mortality due to suicide increased [12,13,14,15,16,17], while mortality due to traffic injuries decreased [12,13,14,17]. In Spain, mortality continued to decrease at the start of the crisis for most causes [18].

Financial crises usually affect the most deprived populations because they have a higher probability of job and even housing loss, given the high number of evictions in Spain during the crisis [19]. Studies analyzing the effects of the financial crisis on inequalities in health have reported heterogeneous results. Inequalities have increased in some countries, but have remained stable in others [4,12,20]. In Spain, the crisis has had the strongest impact in the most disadvantaged population groups and inequalities in health have increased, depending on the health indicator analyzed [7,21,22].

Few studies have analyzed the impact of the financial crisis on health at the city level, where socioeconomic inequalities in mortality in small areas of cities have been described, with the most disadvantaged areas showing the highest mortality [23,24,25,26,27,28]. Local governments in Spain have no responsibility in the management of healthcare services; however, they are responsible for fundamental aspects of social and health protection that can improve the health of the population, and especially, reduce health inequalities.

Thus, the aim of this study was to analyze the trend in socio-economic inequalities in mortality in small areas due to several specific causes before (2001–2004, 2005–2008) and during (2009–2012) the economic crisis in seven Spanish cities.

## 2. Methods

### 2.1. Design, Unit of Analysis, and Study Population

This ecological study of trends was based on three periods (2001–2004, 2005–2008, and 2009–2012) and forms part of the IMCRISES project (the effect of the economic crisis on mortality, reproductive health, and health inequalities in Spain) [29]. The units of analysis were the census tracts of seven Spanish cities, as defined in the 2001 Spanish Population and Housing Census. These cities vary in size and are located in different geographical regions of Spain (Center, South, North, and East). The study population consisted of the residents of the seven cities during the period 2001–2012.

### 2.2. Information Sources

Mortality data grouped by sex, census tract, and period were obtained from the mortality registers of the corresponding cities or autonomous communities (Spanish regions). The census tract was obtained from the postal address of the deceased provided on the death certificate or from the Register of Inhabitants of each city. Population data stratified by age (in 5-year groups), sex, census tract, and period were obtained from the Register of Inhabitants for each city or from the National Institute of Statistics (Instituto Nacional de Estadística). The 2001 Population and Housing Census was used to obtain the information needed to construct the socioeconomic deprivation index for each city [30].

### 2.3. Mortality

This study analyzed all-cause mortality and deaths due to several specific causes, namely, infectious and parasitic diseases, diabetes mellitus, ischemic heart disease, cerebrovascular diseases, cirrhosis, suicide, and traffic injury. These causes of death are those most strongly affected by economic crises [4,8,12,20]. The underlying cause of death was coded using the International Classification of Diseases, tenth revision (ICD-10) (see the codes in the footnote to Table 1).

### 2.4. Socioeconomic Deprivation Index

We included a socioeconomic deprivation index as a covariate, which was the first component of a principal components analysis performed within each city for five socioeconomic indicators corresponding to 2001 and available for each census tract, based on the methodology described by Domínguez-Berjón et al. [31]. The indicators included in the index were the percentages of: (a) Unemployment: percentage of people ≥16 years or over who were unemployed or actively seeking a job in relation to the total economically active population; (b) low educational level (primary school or lower): percentage of people ≥16 years with less than 5 years of schooling or with 5 years of schooling or more who did not complete basic compulsory education, in relation to the total population ≥16 years; (c) low educational level in young people (16–29 years); (d) manual workers: percentage of people ≥16 years, in manual employment in relation to the total employed population ≥16 years; and (e) temporary workers: Percentage of people ≥16 years, employed in temporary jobs, in relation to the total employed population ≥16 years. A socioeconomic deprivation index was obtained for each city. Each index was normalized to achieve a mean of 0 and standard deviation of 1. In all cities, the index accounted for over 75% of the variability of the included indicators.

### 2.5. Data Analysis

Age-standardized mortality rates (ASMR) were calculated by the direct method using the Spanish population for the year 2001 as the reference population. ASMR were calculated for each cause of death, sex, period, and city.

For small areas, the mortality indicator used for the analysis was the standardized mortality ratio (SMR). The SMR is dependent on population size since its variance is inversely proportional to the expected values; thus, areas with a small population tend to show highly variable estimates. To smooth the SMR, we used the hierarchical Bayesian model proposed by Besag, York, and Mollié [32]. This model takes two types of random effects into account, spatial and heterogeneous: the former takes account of the spatial structure of the data, while the latter deals with non-structural (non-spatial) variability. Smoothed SMR (sSMR) were estimated for each cause of death, period, and city with the following model:*O_i_*~*Poisson* (*E_i_θ_i_*)log(*θ_i_*) = α + *S_i_* + *H_i_* (model 1)
where, for each area *i*, *O_i_* is the number of observed cases, *E_i_* the expected cases, *θ_i_* the sSMR with respect to the whole city, *S_i_* the spatial effect, and *H_i_* the heterogeneous effect. The expected cases were calculated by indirect standardization taking as reference the mortality rates of each city by age (in five-year groups), cause of death, and period. 

The geographical distribution of the sSMR is represented using maps of septiles. The deprivation index results are also represented as septile maps. All maps were generated using the R statistical package [33].

To analyze the relationship between socioeconomic deprivation and mortality in the three periods, we fitted an ecological regression model for each city that included the deprivation index (D), the period (though two dummy variables *P*_2_ and *P*_3_), and their interaction:*O_it_*~*Poisson* (*E_it_θ_it_*)log(*θ_it_*) = α + *β*_1_*D*_1_ + *β*_2_*P*_2*t*_ + *β*_3_*P*_3*t*_ + *β*_4_*P*_2*t*_*D_i_* + *β*_5_*P*_3*t*_*D_t_* + *S_it_* + *H_it_* (model 2)
where, for each area *i* and period *t* (*t* = 1 for period 2001–2004, *t* = 2 for period 2005–2008 and *t* = 3 for period 2009–2012), *O_it_* is the number of observed cases, *E_it_* the expected cases, *θ_it_* the sSMR with respect to the whole city, *S_it_* the spatial effect, and *H_it_* the heterogeneous effect. Finally, two dummy variables *P*_2*t*_ and *P*_3*t*_ take the following values: *P_jt_* = 1 if *j* = t and *P_jt_* = 0 otherwise. The expected cases were calculated taking as reference the mortality rates of the first period (2001–2004). Thus, the risk of mortality associated with the deprivation index was calculated as exp(*β*_1_) in the first period, as exp(*β*_1_ + *β*_4_) in the second period, and as exp(*β*_1_ + *β*_5_) in the third period. Changes between periods in the relationship between the socioeconomic deprivation index and mortality were evaluated through the interactions included in model 2. Specifically, we studied the change between the first and second period (*β*4) and the second and third periods (*β*5–*β*4).

In the two models, an intrinsic conditional autoregressive prior distribution (ICAR) [32] was assigned to the spatial effect, which assumed that the expected value of each area coincided with the mean of the spatial effect of the adjacent areas and had a variance of $\sigma_{\mathrm{Sr}}^{2}$, while the heterogeneous effect was represented by using a normal distribution with mean 0 and variance $\sigma_{\mathrm{Hr}}^{2}$. A half-normal distribution with mean 0 and precision 0.001 was assigned to the standard deviation *σ*_Sr_ and *σ*_H_. A normal vague prior distribution was assigned to the parameters α, β_1_, …, β_5_.

As the deprivation index scale is dimensionless and arbitrarily fixed, for each cause of death and city, we calculated the increase in risk corresponding to a change in deprivation index from its 95th percentile value (P_95_) (severe deprivation) to its 5th percentile value (P_5_) (low deprivation). 

Relative risk (RR) estimates were obtained based on the mean of their posterior distribution, along with the corresponding 95% credible intervals (95%CI). All distributions were obtained using the “Integrated nested Laplace approximations” method (INLA), using the INLA library of the R statistical package, R.3.5.0. [34].

Finally, estimation of RRs for suicide and traffic injuries was not feasible for San Sebastian and Granada due to the small number of deaths among women.

## 3. Results

Table 1 and Table 2 show, for each city and period the population, the number of deaths and the age-standardized mortality rate (ASMR) by cause of death for men and women, respectively. In general, the all-cause and cause-specific mortality rates decreased in the second and third periods in most of the cities in both men and women. For example, in Madrid, in the first period, the ASMR for all causes was 1112.1 per 100,000 inhabitants, which fell to 883.9 in the third period among men. In women, the rates were 581.0 in the first period and 492.0 in the third.

In general, the geographical pattern was similar for the socioeconomic deprivation index and sSMR. For example, the Figure 1 shows the geographical distribution of the socioeconomic deprivation index and the distribution, in the three periods, of sSMR for cirrhosis among men in Barcelona and of sSMR for ischemic heart disease among women in Sevilla. Green areas represent low sSMR and low socioeconomic deprivation, while brown areas represent high sSMR and high socioeconomic deprivation. 

Figure 2 and Table 3 show the association between the deprivation index and all-cause and cause-specific mortality in the three periods in men. In all cities, there was an association between the deprivation index and all-cause mortality that remained stable over the three periods. Regarding specific causes, in most cities there was a positive association between the deprivation index and mortality due to infectious and parasitic diseases that remained stable over the three periods. The same pattern was observed for mortality from cirrhosis, except in San Sebastian, where there was no association in the first (RR = 1.55, 95%CI = 0.52–3.39) and second (RR = 1.69, 95%CI = 0.75–3.31) periods but an association appeared in the third when residents of the most deprived census tracts were 6.49 times more likely to die of cirrhosis (RR = 6.49, 95%CI = 2.29–15.86) and this increase was significant. In relation to ischemic heart disease, an association was found between deprivation and mortality in Barcelona and Madrid that remained stable over time, and an association appeared in the third period when the risk of mortality from ischemic heart diseases was 1.85 (RR = 1.85, 95%CI = 1.19–2.77) in Granada and 1.39 in (RR=1.39, 95%CI = 1.06–1.80) Sevilla. In addition, there was an association between deprivation and mortality due to cerebrovascular disease in the second and third periods in Barcelona, while in Vitoria there was no association in the first (RR = 1.39, 95%CI = 0.84–2.16) and second (RR = 0.91, 95%CI = 0.52–1.47) periods; however, in the third period, the residents of the most-deprived census tract were 1.91 times more likely to die of cerebrovascular diseases (RR = 1.91, 95%CI = 1.18–2.93) and this increase was significant. An association was found between deprivation and mortality due to suicide and that remained stable over the three periods only in Sevilla and there was an association in the third period in Madrid and San Sebastian. An association with diabetes appeared in the third period in Barcelona, Bilbao, Madrid, and Granada. Finally, most of the cities showed no association between deprivation and mortality due to traffic injuries in any of the periods.

Figure 3 and Table 4 show the association between the deprivation index and cause-specific and all-cause mortality in the three periods in women. In Barcelona, Granada, and Sevilla, residents of most-deprived census tracts were more likely to die in the third period (which was significant for Barcelona and Granada). Regarding specific causes, in Barcelona, Madrid, and Sevilla, there was a positive association between the deprivation index and mortality due to infectious and parasitic diseases that remained stable over the three periods. The same pattern was observed for mortality due to diabetes, except there was no association in the first period in Sevilla, and in Granada there was no association in the second (RR = 1.75, 95%CI = 0.61–3.82) period but an association appeared in the third, when the residents of most-deprived census tracts were 4.20 times more likely to die of diabetes (RR = 4.20, 95%CI = 1.77–8.46). In relation to ischemic heart disease and cirrhosis, there was association between the deprivation index and mortality in Barcelona and Sevilla that remained stable over time. In addition, in Bilbao, there was no association between the index and mortality due to ischemic heart disease in the first (RR = 1.17, 95%CI = 0.81–1.64) and second (RR = 1.27, 95%CI = 0.78–1.92) periods but an association appeared in the third period with a RR of 1.49 (RR = 1.49, 95%CI = 1.05–2.04). In addition, in most cities, there was no association between deprivation and mortality due to cerebrovascular diseases except in Granada, where there was association in the second and third periods. Finally, most of the cities showed no association between deprivation and mortality due to suicide and traffic injuries in any of the periods.

## 4. Discussion

### 4.1. Main Findings of the Study

In all cities, in both men and women, mortality rates for most of the causes analyzed tended to decrease in the three periods studied, including the period of the financial crisis. In general, socioeconomic inequalities were found for many of the causes analyzed, especially in men. Among women, inequalities were less important. In some cities, and for some specific causes of death, previously absent inequalities appeared in the crisis period, especially among men. This suggests a possible increase in mortality inequalities for some specific causes associated with the crisis, and perhaps, additionally associated with specific local contexts.

### 4.2. What Is Already Known and What Does This Study Add?

Studies conducted in Europe have found that the crisis was associated with a decrease in all-cause mortality and mortality from certain specific causes, such as cardiovascular disease, cirrhosis, infectious diseases, and traffic injuries [12,13,14,18,20], and with an increase in mortality due to suicide [12,13,14,15,16,20,35,36,37,38]. Our results are in line with those of these studies, except for mortality due to suicide, which in this study continued to decrease after the start of the financial crisis. Studies performed in Spain in the first years of the crisis [18] have reported the same decreasing tendency. However, more recent reports have shown that mortality due to suicide increased after the start of the crisis, especially in men [39,40]. Moreover, two studies based on urban areas did not show an increase in inequalities in suicide mortality during the crisis period [41,42].

Of note, the results found can be partially explained by the different incubation periods of the diseases. Suicide mortality can be an effect of the job loss and financial strain due to the economic crisis [11,12]. Infectious diseases also have a short incubation period and their incidence has risen in some countries during the crisis [7], but in our study this is not reflected in mortality data. Other relevant factors are the mechanisms that explain the changes in health outcomes during the crisis period for each cause of mortality. Catalano et al. (2011) [11] explains how the stress mechanism due to unemployment and job loss can increase unhealthy behaviors as a coping measure, such as alcohol intake or other drug use, which can be related to some diseases.

Our study demonstrates the existence of socioeconomic inequalities in all-cause mortality that remained stable throughout the study period in men and increased in two cities among women. A study conducted in Finland in the 1990s reported that mortality followed the same tendency after the start of the financial crisis [41]. However, several studies have reported that inequalities increased in periods of economic crisis, as we found for women in Barcelona and Granada [4,24,43].

Previous studies have detected socioeconomic inequalities in mortality due to ischemic heart disease and cerebrovascular disease in periods prior to the onset of the crisis in small areas in various European cities [25]. Some studies have found socioeconomic inequalities in mortality due to cardiovascular disease (ischemic heart disease and cerebrovascular disease) that persisted after the onset of the financial crisis [44,45]. This result is in agreement with our findings for most of the cities in our study.

In periods before the crisis, socioeconomic inequalities in mortality have been found in mortality due to traffic injuries in small areas of several Spanish [28] and European [46] cities, especially in men. In agreement with our results, subsequent studies have found no socioeconomic inequalities due to this cause after the start of the crisis [45].

We have found no studies analyzing the effects of the financial crisis on socioeconomic inequalities in mortality due to infectious diseases, diabetes, or cirrhosis. However, some studies have reported the existence of these inequalities in periods before the onset of the crisis [25,27].

Most causes of death on social inequalities in mortality seem to differ in men and women, being less important in women. This could be explained by the distinct distribution of risk factors between men and women as a result of socially accepted differences in gender-related roles and powers. Traditional gender roles produce greater pressure on men in terms of earnings and unemployment and mainly on those from less advantageous social classes. Moreover, hegemonic masculinity is more closely related to risky health behaviors [47,48]. Total mortality increased in Barcelona and Granada among women during the economic crisis, although death rates and the increase in the risk of death in the third period were much higher in Granada, a city located in the south of Spain where the crisis has been more keenly experienced [49].

As shown, it is not easy to quantify the effects of the financial crisis on mortality and it is even more difficult to quantify those of socioeconomic inequalities in mortality. Moreover, the results of this study and those in the literature are inconclusive and sometimes contradictory. Some of the factors that could explain the discrepant results are the following: the heterogeneity of the socioeconomic variables analyzed, the type of measures used to analyze the magnitude of social inequalities in health, and the various state social policies implemented during the crisis [4]. In Spain, the contribution of social policies and austerity on health results is still controversial [4,21,45,50,51,52]. It is worth mentioning that Spain has a national health system that can protect the treatment of population diseases. This system suffered from austerity measures, but in our study, it is difficult to discern the possible influence of health budget cuts on the increase in inequalities, given that our data were gathered until 2012 and these policies only began to be implemented in 2011. However, the effect of the economic crisis in Spain has been heterogeneous in the different autonomous communities, which may be because each region has some control over how socials cuts are implemented [4,7]. Another important factor is the diversity of time periods studied. The use of short periods for the analyses makes it difficult to capture the long-term effects on mortality inequalities. It is reasonable to expect that the effects of socioeconomic circumstances on mortality will not be contemporaneous but will become apparent in short and long-term effects. Finally, another factor is the changes in the socioeconomic groups compared during the periods. In Spain, since 2000, the foreign-born population has increased, although this migratory flow decreased after the beginning of the financial crisis [53]. The growth of immigration could present residential segregation patterns because immigrants from low-income countries tend to increase in socioeconomically disadvantaged neighborhoods [54]. In this regard, some studies have found that the increase in the foreign-born population could explain the decrease in mortality inequalities between more and less-disadvantaged areas [53,55].

### 4.3. Limitations of This Study

The main limitation of the present study is the use of a single deprivation index in the three periods studied. Although the socio-economic situation may have changed over the period studied, we believe that the distribution of deprivation at the small area level has not undergone substantial variations over time because the processes involving changes in socioeconomic deprivation at area level are slow [56]. To provide information on the influence of the financial crisis on socioeconomic inequalities in health, this study analyzed the possible effect of the crisis on socioeconomic inequalities in various causes of mortality in seven Spanish cities with different economic contexts. Moreover, until 2012, mortality statistics in Spain did not incorporate information on the educational attainment of the deceased, and consequently ecological studies were needed to analyze socioeconomic inequalities in mortality. In this context, it is important to stress that small area studies are those that most closely approximate individual-level studies and are least susceptible to the component of ecological bias created by heterogeneity within areas of exposure or other determining factors [57]. Moreover, they allow detection of geographical patterns in mortality and deprivation that might not be evident in larger geographical areas. Finally, in some cities, it was not possible to analyze all specific causes due to the small numbers of deaths observed. The low number of deaths may have led to the low statistical power of the data to detect significant associations. However, the statistical method chosen is intended to reduce the impact this may have on the results.

## 5. Conclusions

This study shows that socioeconomic inequalities in mortality in small urban areas remained stable after the onset of the financial crisis for all-cause mortality, especially in men. Among women, inequalities were less important. In some cities, and for some specific causes of death, inequalities appeared in the crisis period, which had been absent in the previous periods, especially among men. In addition, there is evidence that economic crises have an impact on some health determinants such as the deterioration of the labor market, the difficulty of accessing housing, worse health behaviors, etc. In future, it will be necessary to continue to monitor the population’s health and its determinants, and invest in information systems that enhance knowledge of the effect of economic crises on the health and wellbeing of the population. Moreover, further studies are needed on the effects of economic crises on health and inequalities in health, since multiple indicators can reflect negative effects in the long-term. Finally, it is important to stress that the policy response in managing the financial crises can mitigate or magnify the negative impacts on health and health inequalities. Therefore, financial crises provide a good opportunity to mainstream health in all policies and promote intersectional policies to improve the health of the population.

## Figures and Tables

**Figure 1 ijerph-17-00958-f001:**
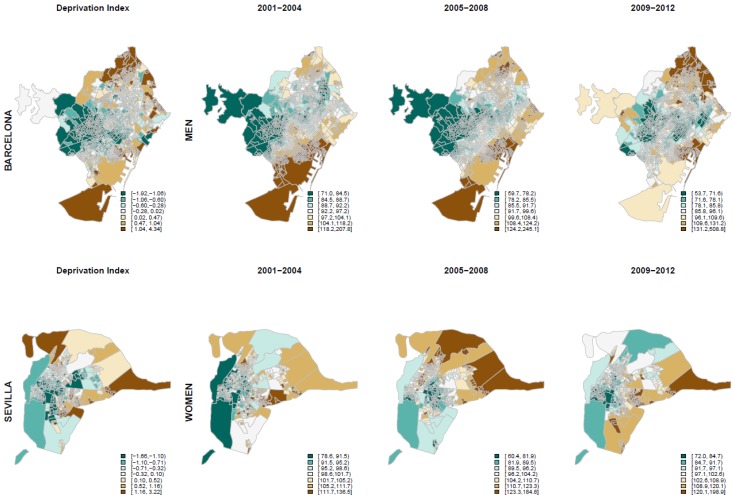
Deprivation index and Smootherd Standardized Mortality Ratios for the 3 periods among men in Barcelona and women in Sevilla.

**Figure 2 ijerph-17-00958-f002:**
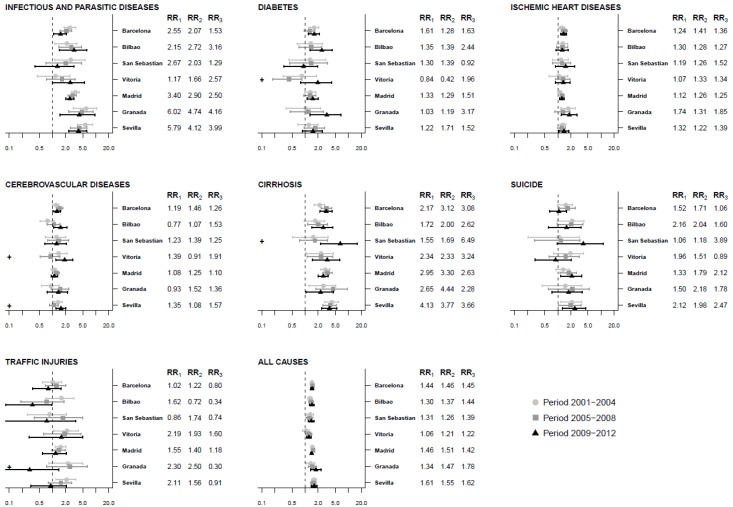
Association between the deprivation index and mortality. Relative risk comparing the 95th to 5th percentiles of the deprivation index by period and cause of death in seven Spanish cities for men, 2001–2012. Note: **+** indicates that RR_3_ is statistically significantly different than RR_2_. RR1: RR 2001–2004 period, RR2: RR 2005–2008 period, RR3: RR 2009–2012 period.

**Figure 3 ijerph-17-00958-f003:**
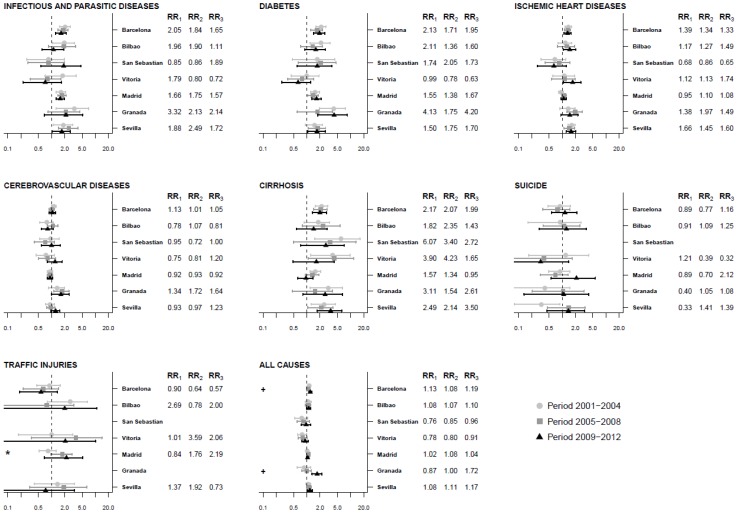
Association between the deprivation index and mortality. Relative risk comparing the 95th to 5th percentiles of the deprivation index by period and cause of death in seven Spanish cities for women, 2001–2012. Note: * indicates that RR_2_ is statistically significantly different than RR_1_ and **+** indicates that RR_3_ is statistically significantly different than RR_2_. RR1: RR 2001–2004 period, RR2: RR 2005–2008 period, RR3: RR 2009–2012 period.

**Table 1 ijerph-17-00958-t001:** Population, number of deaths (N), and age-standardized mortality rate (ASMR) per 100,000 inhabitants by period and cause of death in 7 Spanish cities for men, 2001–2012.

			Infectious and Parasitic Diseases		Diabetes		Ischemic Heart Diseases		Cerebrovascular Diseases		Cirrhosis		Suicide		Traffic Injuries		All Causes	
Cities (No. Census Tract)	Period	Pob	N	ASMR	N	ASMR	N	ASMR	N	ASMR	N	ASMR	N	ASMR	N	ASMR	N	ASMR
**Barcelona** (1491)	2001–2004	293,6440	799	26.9	725	25.0	3565	122.3	2187	77.4	672	21.8	342	11.4	285	9.7	32,128	1105.9
	2005–2008	304,6533	768	24.3	702	22.6	3047	98.5	1957	63.8	553	17.4	339	10.6	195	6.5	31,247	1011.1
	2009–2012	308,3469	609	18.1	648	18.9	2835	84.4	1658	48.6	557	17.1	332	10.2	134	4.3	29,724	883.9
**Bilbao** (288)	2001–2004	657,840	174	26.5	129	21.7	698	113.0	549	91.4	174	25.0	78	11.5	84	13.0	7282	1177.0
	2005–2008	658,272	177	25.0	131	19.0	710	101.5	530	77.9	161	22.2	82	11.5	46	7.0	7350	1065.5
	2009–2012	660,882	119	15.1	145	18.7	729	94.0	470	61.3	152	19.5	69	9.8	43	6.3	7392	963.3
**San Sebastian** (140)	2001–2004	333,822	81	26.1	66	23.7	318	106.5	226	79.3	76	22.8	37	11.3	53	16.1	3346	1131.8
	2005–2008	336,108	68	19.9	63	18.9	283	84.4	217	66.5	88	23.6	42	11.8	38	11.0	3419	1018.2
	2009–2012	339,319	51	13.6	82	22.6	271	72.1	221	60.0	59	15.0	34	9.6	17	5.1	3353	904.6
**Vitoria** (168)	2001–2004	429,826	89	25.5	58	19.4	282	88.3	233	80.4	84	21.7	59	14.4	86	20.9	3426	1074.5
	2005–2008	451,026	82	21.4	84	23.3	288	75.0	227	63.1	100	23.5	66	15.1	49	11.4	3680	971.6
	2009–2012	468,668	59	12.9	75	17.6	306	68.9	206	48.1	94	20.0	62	12.6	24	5.1	3808	864.0
**Madrid** (2358)	2001–2004	5,498,348	1235	23.4	664	14.3	5788	119.4	3037	64.9	1056	19.8	432	7.9	554	9.9	54,053	1112.1
	2005–2008	5,668,478	1132	20.1	595	11.6	5262	100.2	2810	54.4	915	16.4	240	4.1	355	6.2	53,087	1008.3
	2009–2012	5,776,884	1098	18.3	649	11.2	4427	76.6	2349	40.7	728	12.4	126	2.1	107	1.8	50,979	883.9
**Granada** (181)	2001–2004	441,668	114	28.4	78	21.3	599	164.2	348	99.2	130	31.9	56	13.2	69	15.0	4285	1169.1
	2005–2008	439,300	118	28.3	90	23.5	509	132.0	345	91.2	143	34.1	50	11.7	53	12.0	4409	1128.9
	2009–2012	446,516	72	15.8	103	23.7	536	122.1	298	69.2	101	22.6	88	19.6	22	4.9	4217	969.3
**Sevilla** (510)	2001–2004	1,357,036	292	24.3	169	17.9	1808	188.6	1398	161.7	321	27.3	145	11.5	154	11.1	11,655	1193.8
	2005–2008	1,332,728	318	26.4	182	18.1	1463	141.9	1309	137.0	319	26.1	130	10.2	104	7.8	12,017	1155.8
	2009–2012	1,337,716	260	20.8	193	18.1	1242	110.7	1002	94.6	323	25.1	115	9.0	56	4.3	11,836	1051.5

International Classification of Diseases (ICD-10): Infectious and parasitic diseases (A00–B99, R75); Diabetes (E10–E14); Ischemic heart diseases (I20–I25), Cerebrovascular diseases (I60–I69), Cirrhosis (K70, K72.1, K73, K74, K76(.1.9)); Suicide (X60–X84); Traffic injuries (V02–V04(.1.9), V09(.2.3), V12–V14(.3.4.5.9), V19(.4.5.6.9), V20–V28(.3.4.5.9), V29–V79(.4.5.6.7.8.9), V80(.3.4.5), V81–V82(.1), V83–V86(.0.1.2.3), V87(.0.1.2.3.4.5.6.7.8), V89(.2.9)).

**Table 2 ijerph-17-00958-t002:** Population, number of deaths (N), and age-standardized mortality rate (ASMR) per 100,000 inhabitants by period and cause of death in 7 Spanish cities for women, 2001–2012.

			Infectious and Parasitic Diseases		Diabetes		Ischemic Heart Diseases		Cerebrovascular Diseases		Cirrhosis		Suicide		Traffic Injuries		All Causes	
Cities (No. Census Tract)	Period	Pob	N	ASMR	N	ASMR	N	ASMR	N	ASMR	N	ASMR	N	ASMR	N	ASMR	N	ASMR
**Barcelona (1491)**	2001–2004	3,298,056	723	15.2	959	17.0	2878	49.8	3441	59.0	443	9.6	163	4.4	146	4.0	33,501	614.0
	2005–2008	3,375,674	749	14.0	885	14.2	2336	37.2	2949	46.1	409	8.7	157	4.1	67	1.8	32,914	557.0
	2009–2012	3,406,659	655	10.9	789	11.5	2102	29.8	2510	35.7	381	7.8	159	4.3	62	1.5	32,306	501.6
**Bilbao (288)**	2001–2004	730,202	136	14.3	177	15.6	504	44.6	731	64.2	94	9.9	29	3.6	42	5.5	6935	636.7
	2005–2008	734,570	134	11.9	189	14.4	463	35.3	691	52.2	61	6.0	50	6.0	17	1.7	7050	567.0
	2009–2012	738,067	147	10.8	167	10.6	447	29.6	645	41.6	76	7.0	33	3.7	9	1.2	7224	502.4
**San Sebastian (140)**	2001–2004	375,249	68	11.7	89	14.7	254	40.9	399	65.4	30	6.0	29	6.7	23	5.1	3664	619.8
	2005–2008	380,058	70	10.7	72	10.7	224	31.8	361	51.8	26	4.9	8	2.1	13	3.0	3625	538.3
	2009–2012	385,302	50	6.5	93	11.8	191	22.4	308	37.5	35	6.1	18	3.4	9	1.7	3824	504.8
**Vitoria (168)**	2001–2004	443,772	71	14.6	87	17.6	203	40.7	294	58.0	36	7.9	16	3.6	23	5.1	3094	622.6
	2005–2008	463,781	66	11.8	117	18.7	209	34.6	269	43.3	29	5.7	16	3.2	13	2.5	3231	541.2
	2009–2012	482,422	50	6.9	100	12.9	227	28.8	261	35.2	35	6.0	12	2.2	11	2.5	3417	475.9
**Madrid (2358)**	2001–2004	6,273,028	932	11.3	1038	10.9	4949	51.9	4762	49.8	589	7.4	193	2.8	230	3.6	52,832	581.0
	2005–2008	6,437,031	1017	11.1	901	8.6	4532	43.2	4266	40.7	405	4.9	95	1.4	141	2.1	54,214	544.5
	2009–2012	6,599,904	1045	10.0	983	8.1	3934	33.0	3758	31.5	361	4.0	34	0.5	41	0.6	54,534	492.0
**Granada (181)**	2001–2004	508,984	63	10.7	102	16.3	473	74.8	481	74.6	76	13.1	27	5.1	19	3.6	4206	678.1
	2005–2008	505,528	93	14.2	103	14.7	435	60.7	499	68.9	54	8.6	36	6.6	10	2.0	4547	658.1
	2009–2012	513,880	60	8.0	113	14.3	574	68.3	430	51.8	52	8.1	27	4.9	6	1.3	4561	581.9
**Sevilla (510)**	2001–2004	1,482,864	181	11.3	270	15.9	1457	85.6	2583	147.7	112	7.2	48	3.1	45	3.1	11,545	684.0
	2005–2008	1,463,852	188	11.0	310	16.8	1101	59.9	2207	115.7	115	7.1	51	3.3	23	1.6	11,805	650.7
	2009–2012	1,474,368	194	10.3	232	10.8	1078	52.7	1583	74.1	115	6.6	56	3.7	14	1.0	12,056	599.6

**Table 3 ijerph-17-00958-t003:** Association between mortality and the deprivation index. Relative risk comparing the 95th to 5th percentiles of the deprivation index by period and cause of death in 7 Spanish cities for men, 2001–2012.

		Infectious and Parasitic Diseases		Diabetes		Ischemic Heart Diseases		Cerebrovascular Diseases		Cirrhosis		Suicide		Traffic Injuries		All Causes	
Cities (No. Census Tract)	Period	RR	95%CI	RR	95%CI	RR	95%CI	RR	95%CI	RR	95%CI	RR	95%CI	RR	95%CI	RR	95%CI
**Barcelona** (1491)	2001–2004	2.55	(1.85–3.41)	1.61	(1.23–2.05)	1.24	(1.08–1.44)	1.19	(0.99–1.41)	2.17	(1.60–2.86)	1.52	(1.00–2.19)	1.02	(0.59–1.63)	1.44	(1.36–1.53)
	2005–2008	2.07	(1.51–2.79)	1.28	(0.96–1.66)	1.41	(1.22–1.64)	1.46	(1.22–1.73)	3.12	(2.31–4.11)	1.71	(1.11–2.54)	1.22	(0.68–2.02)	1.46	(1.37–1.55)
	2009–2012	1.53	(1.07–2.10)	1.63	(1.19–2.16)	1.36	(1.16–1.59)	1.26	(1.02–1.55)	3.08	(2.18–4.16)	1.06	(0.66–1.59)	0.80	(0.35–1.56)	1.45	(1.35–1.55)
**Bilbao** (288)	2001–2004	2.15	(1.13–3.67)	1.35	(0.65–2.46)	1.30	(0.97–1.72)	0.77	(0.53–1.10)	1.72	(0.95–2.85)	2.16	(0.98–4.05)	1.62	(0.70–3.13)	1.30	(1.13–1.48)
	2005–2008	2.72	(1.41–4.64)	1.39	(0.72–2.45)	1.28	(0.94–1.70)	1.07	(0.76–1.49)	2.00	(1.15–3.26)	2.04	(0.96–3.79)	0.72	(0.19–1.86)	1.37	(1.21–1.54)
	2009–2012	3.16	(1.46–6.12)	2.44	(1.31–4.17)	1.27	(0.90–1.74)	1.53	(1.06–2.16)	2.62	(1.43–4.35)	1.60	(0.60–3.48)	0.34	(0.08–0.96)	1.44	(1.27–1.63)
**San Sebastian** (140)	2001–2004	2.67	(1.11–5.61)	1.30	(0.42–2.97)	1.19	(0.76–1.75)	1.23	(0.69–2.01)	1.55	(0.52–3.39)	1.06	(0.21–3.15)	0.86	(0.26–2.09)	1.31	(1.07–1.58)
	2005–2008	2.03	(0.71–4.67)	1.39	(0.44–3.30)	1.26	(0.77–1.96)	1.39	(0.77–2.34)	1.69	(0.75–3.31)	1.18	(0.29–3.27)	1.74	(0.38–5.02)	1.26	(1.05–1.50)
	2009–2012	1.29	(0.40–3.09)	0.92	(0.32–1.97)	1.52	(0.89–2.44)	1.25	(0.66–2.11)	6.49	(2.29–15.86)	3.89	(0.98–11.31)	0.74	(0.04–3.45)	1.39	(1.19–1.62)
**Vitoria** (168)	2001–2004	1.17	(0.44–2.51)	0.84	(0.31–1.83)	1.07	(0.63–1.68)	1.39	(0.84–2.16)	2.34	(1.05–4.54)	1.96	(0.81–3.95)	2.19	(1.07–3.95)	1.06	(0.87–1.26)
	2005–2008	1.66	(0.72–3.30)	0.42	(0.18–0.85)	1.33	(0.87–1.94)	0.91	(0.52–1.47)	2.33	(1.11–4.39)	1.51	(0.63–3.10)	1.93	(0.62–4.50)	1.21	(1.02–1.43)
	2009–2012	2.57	(1.00–5.52)	1.96	(0.78–4.14)	1.34	(0.89–1.94)	1.91	(1.18–2.93)	3.24	(1.48–6.26)	0.89	(0.30–2.07)	1.60	(0.29–5.00)	1.22	(1.07–1.39)
**Madrid** (2358)	2001–2004	3.40	(2.76–4.16)	1.33	(0.98–1.78)	1.12	(1.00–1.26)	1.08	(0.94–1.23)	2.95	(2.30–3.72)	1.33	(0.91–1.89)	1.55	(1.16–2.03)	1.46	(1.39–1.53)
	2005–2008	2.90	(2.32–3.59)	1.29	(0.91–1.77)	1.26	(1.12–1.42)	1.25	(1.08–1.43)	3.30	(2.58–4.18)	1.79	(1.15–2.67)	1.40	(0.98–1.95)	1.51	(1.44–1.58)
	2009–2012	2.50	(2.01–3.09)	1.51	(1.12–1.99)	1.25	(1.10–1.41)	1.10	(0.93–1.30)	2.63	(2.02–3.38)	2.12	(1.15–3.60)	1.18	(0.59–2.09)	1.42	(1.35–1.49)
**Granada** (181)	2001–2004	6.02	(3.08–10.47)	1.03	(0.36–2.25)	1.74	(1.15–2.51)	0.93	(0.54–1.44)	2.65	(1.23–5.06)	1.50	(0.43–3.61)	2.30	(0.83–5.01)	1.34	(1.05–1.66)
	2005–2008	4.74	(2.37–8.67)	1.19	(0.45–2.61)	1.31	(0.86–1.89)	1.52	(0.93–2.37)	4.44	(1.78–9.50)	2.18	(0.61–5.36)	2.50	(0.74–6.48)	1.47	(1.19–1.80)
	2009–2012	4.16	(1.48–9.56)	3.17	(1.31–6.69)	1.85	(1.19–2.77)	1.36	(0.76–2.25)	2.28	(1.03–4.43)	1.78	(0.74–3.64)	0.30	(0.01–1.41)	1.78	(1.33–2.33)
**Sevilla** (510)	2001–2004	5.79	(3.76–8.58)	1.22	(0.67–2.00)	1.32	(1.13–1.54)	1.35	(1.05–1.70)	4.13	(2.82–5.87)	2.12	(1.18–3.55)	2.11	(1.19–3.49)	1.61	(1.44–1.81)
	2005–2008	4.12	(2.62–6.19)	1.71	(0.95–2.85)	1.22	(0.98–1.51)	1.08	(0.85–1.34)	3.77	(2.60–5.32)	1.98	(1.05–3.46)	1.56	(0.81–2.74)	1.55	(1.36–1.76)
	2009–2012	3.99	(2.39–6.29)	1.52	(0.86–2.50)	1.39	(1.06–1.80)	1.57	(1.21–2.02)	3.66	(2.38–5.38)	2.47	(1.27–4.37)	0.91	(0.31–2.09)	1.62	(1.38–1.89)

**Table 4 ijerph-17-00958-t004:** Association between mortality and the deprivation index. Relative risk comparing the 95th to 5th percentiles of the deprivation index by period and cause of death in 7 Spanish cities for women, 2001–2012.

		Infectious and Parasitic Diseases		Diabetes		Ischemic Heart Diseases		Cerebrovascular Diseases		Cirrhosis		Suicide		Traffic Injuries		All Causes	
Cities (No. Census Tract)	Period	RR	95%CI	RR	95%CI	RR	95%CI	RR	95%CI	RR	95%CI	RR	95%CI	RR	95%CI	RR	95%CI
**Barcelona** (1491)	2001–2004	2.05	(1.54–2.66)	2.13	(1.63–2.70)	1.39	(1.17–1.62)	1.13	(0.99–1.28)	2.17	(1.54–2.97)	0.89	(0.47–1.52)	0.90	(0.46–1.56)	1.13	(1.06–1.20)
	2005–2008	1.84	(1.40–2.40)	1.71	(1.24–2.25)	1.34	(1.14–1.56)	1.01	(0.87–1.17)	2.07	(1.43–2.89)	0.77	(0.36–1.48)	0.64	(0.23–1.45)	1.08	(1.00–1.15)
	2009–2012	1.65	(1.23–2.17)	1.95	(1.42–2.56)	1.33	(1.10–1.61)	1.05	(0.88–1.24)	1.99	(1.36–2.81)	1.16	(0.56–2.18)	0.57	(0.18–1.37)	1.19	(1.10–1.29)
**Bilbao** (288)	2001–2004	1.96	(1.06–3.33)	2.11	(1.20–3.39)	1.17	(0.81–1.64)	0.78	(0.54–1.09)	1.82	(0.87–3.33)	0.91	(0.17–2.69)	2.69	(0.79–6.54)	1.08	(0.91–1.25)
	2005–2008	1.90	(0.85–3.52)	1.36	(0.72–2.30)	1.27	(0.78–1.92)	1.07	(0.81–1.40)	2.35	(0.80–5.70)	1.09	(0.37–2.56)	0.78	(0.07–3.20)	1.07	(0.91–1.24)
	2009–2012	1.11	(0.60–1.89)	1.60	(0.90–2.65)	1.49	(1.05–2.04)	0.81	(0.59–1.09)	1.43	(0.59–2.94)	1.25	(0.29–3.52)	2.00	(0.03–10.87)	1.10	(0.97–1.24)
**San Sebastian** (140)	2001–2004	0.85	(0.27–1.97)	1.74	(0.47–4.28)	0.68	(0.33–1.21)	0.95	(0.54–1.50)	6.07	(1.50–16.39)					0.76	(0.52–1.06)
	2005–2008	0.86	(0.28–1.98)	2.05	(0.64–4.70)	0.86	(0.41–1.52)	0.72	(0.41–1.16)	3.40	(0.49–13.19)					0.85	(0.60–1.14)
	2009–2012	1.89	(0.57–4.68)	1.73	(0.59–3.78)	0.65	(0.28–1.24)	1.00	(0.58–1.59)	2.72	(0.72–7.19)					0.96	(0.75–1.21)
**Vitoria** (168)	2001–2004	1.79	(0.72–3.68)	0.99	(0.44–1.91)	1.12	(0.61–1.89)	0.75	(0.44–1.15)	3.90	(1.22–9.68)	1.21	(0.17–4.08)	1.01	(0.18–3.07)	0.78	(0.61–0.98)
	2005–2008	0.80	(0.33–1.64)	0.78	(0.37–1.44)	1.13	(0.57–2.00)	0.81	(0.50–1.25)	4.23	(1.06–11.86)	0.39	(0.02–1.71)	3.59	(0.35–14.17)	0.80	(0.63–1.00)
	2009–2012	0.72	(0.22–1.67)	0.63	(0.27–1.22)	1.74	(1.07–2.67)	1.20	(0.79–1.75)	1.65	(0.48–4.19)	0.32	(0.02–1.39)	2.06	(0.08–10.08)	0.91	(0.77–1.07)
**Madrid** (2358)	2001–2004	1.66	(1.28–2.11)	1.55	(1.22–1.93)	0.95	(0.83–1.07)	0.92	(0.81–1.05)	1.57	(1.17–2.07)	0.89	(0.49–1.50)	0.84	(0.51–1.27)	1.02	(0.97–1.08)
	2005–2008	1.75	(1.39–2.17)	1.38	(1.04–1.80)	1.10	(0.96–1.25)	0.93	(0.81–1.07)	1.34	(0.92–1.88)	0.70	(0.32–1.35)	1.76	(0.95–2.97)	1.08	(1.03–1.14)
	2009–2012	1.57	(1.26–1.95)	1.67	(1.29–2.11)	1.08	(0.93–1.25)	0.92	(0.80–1.06)	0.95	(0.62–1.38)	2.12	(0.56–5.74)	2.19	(0.69–5.20)	1.04	(0.98–1.10)
**Granada** (181)	2001–2004	3.32	(1.26–7.01)	4.13	(1.97–7.54)	1.38	(0.88–1.99)	1.34	(0.84–1.96)	3.11	(1.19–6.52)	0.40	(0.04–1.48)			0.87	(0.60–1.20)
	2005–2008	2.13	(0.83–4.45)	1.75	(0.61–3.82)	1.97	(1.30–2.88)	1.72	(1.12–2.50)	1.54	(0.47–3.85)	1.05	(0.21–3.17)			1.00	(0.76–1.26)
	2009–2012	2.14	(0.69–5.09)	4.20	(1.77–8.46)	1.49	(0.93–2.28)	1.64	(1.00–2.55)	2.61	(0.81–6.56)	1.08	(0.13–4.06)			1.72	(1.31–2.22)
**Sevilla** (510)	2001–2004	1.88	(1.10–3.00)	1.50	(0.92–2.27)	1.66	(1.36–1.99)	0.93	(0.72–1.17)	2.49	(1.24–4.47)	0.33	(0.08–0.87)	1.37	(0.41–3.38)	1.08	(0.95–1.22)
	2005–2008	2.49	(1.44–4.03)	1.75	(1.07–2.69)	1.45	(1.09–1.91)	0.97	(0.77–1.18)	2.14	(1.05–3.93)	1.41	(0.45–3.32)	1.92	(0.32–6.38)	1.11	(0.97–1.27)
	2009–2012	1.72	(1.03–2.68)	1.70	(1.04–2.64)	1.60	(1.23–2.03)	1.23	(0.96–1.54)	3.50	(1.73–6.27)	1.39	(0.45–3.29)	0.73	(0.05–3.32)	1.17	(1.01–1.35)

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
