# Peer review of "Effect of the Financial Crisis on Socioeconomic Inequalities in Mortality in Small Areas in Seven Spanish Cities"

_ijerph, 2020, doi:10.3390/ijerph17030958_

Round 1

Reviewer 1 Report

This is an important and timely study--to understand how economic crises impact premature mortality rates in Spain.  The authors used high quality datasets of mortality over three time periods and deprivation indices calculated at the first time period.  Below are a few suggested edits for the author(s).

Introduction

Line 37: When discussing the impacts of economic crises on mortality, please refer to “premature mortality”.

2.3. Data

Please provide a citation to justify the causes of death listed as known “causes of death most strongly affected by economic crises”.

2.4. Data analysis

Please check the ? in the equations and their descriptions (lines 103-105). I suspect that the ? should be replaced with a j?

Please report the R statistical package used to estimate the sSMRs.

For the model 2 example, report “…(for this example, two dummy variables P2 and P3) and their interaction”.

Please add Tables 1 and 2 and the Figures 1 and 2 to the paper rather than including them as supplemental materials.

4.3 Limitations

Please talk about the potential of very small numbers of deaths by specific causes after age and sex-standardization (indirect method) and even though this variation was addressed in the statistical models, what impact may that have on the results. 

Discussion

Please talk more about the incubation periods for infectious vs. chronic disease causes of death and how this continuum may impact the results (e.g., the relatively short period of time between infectious disease exposure and death compared to chronic diseases.  Was the economic crisis a tipping point for some with chronic diseases?  It would be helpful for the reader to have a more in-depth understanding of disease processes in relation to economic crises.  Finally, please talk about the health system in Spain and how it may be responding positively to protect health during the economic crisis--given the relatively null findings.

Author Response

We would like to thank the 3 reviewers. Their comments have helped us to improve the manuscript. Below you have the answers to the comments.

Reviewer 1:

Reviewer 2 Report

In this paper, the authors examine trends in the association between socioeconomic deprivation and mortality at the census-tract level in seven Spanish cities, before and during the economic crisis. The paper’s main research question is whether associations between socio-economic deprivation and (all-cause and cause-specific) mortality in small areas changed as the economic crisis unfolded. The authors find that males living in socio-economically disadvantaged census tracts experienced higher all-cause mortality rates before the financial crisis, and that this association remained stable during the crisis. For women, an association between deprivation and mortality emerged only during the crisis, and only in three out of the seven cities.

The analytic approach is well-conceived and appropriate for the question. Though the paper is generally well-written, its largest potential for improvement is its exposition (framing, discussion of past research, and discussion of results). I outline possible ways to strengthen the exposition, as well as a few other suggestions, below.

***

Title: The authors are very careful throughout the paper to write of associations between deprivation and mortality, but the title sounds a little too causal. The paper, which is very interesting, could also benefit from a more “catchy” title.

Abstract: The abstract should mention that “small areas” here refers to census tracts. It should also be made clearer that the deprivation index, like the sSMR, was calculated for each census tract.

Citations: Sentences on lines 39 and 76 should have citations.

Introduction

Motivation for the study. Currently, the motivation for the study is a little ambiguous. The introduction could be reframed and expanded. How do financial crises impact mortality? What do we know so far, and why are these seven cities the ideal testing ground for filling in the gaps of what we don’t know? Background section. There should be a background section. Currently, the authors cite many studies in passing, but do not discuss any in detail. Specifically, a new background section could contain three key components: On mechanisms through which financial crisis are known/theorized to influence mortality. For example, how do people change their behavior during recessions? How does a decrease in financial resources affect the receipt of medical care? How might these answers differ by socio-economic status? On causes of death. The paper doesn’t need to include an explicit set of hypotheses to be tested, but it would be helpful if it gave enough context on the connection between specific causes of death and financial crises so the reader can easily formulate their own ideas about what to expect. Connecting points a and b with census-tract deprivation. How does small area deprivation influence health/mortality?

As a smaller point, it would also be useful to describe in this section or elsewhere why these seven cities were chosen. What do they have in common/what are their differences?

Methods

How might cities’ compositional change over time shape results? For example, did people move to/away from these cities during the crisis? If so, what biases might this introduce? If people moved to cities, would it be the case that the relative risk estimates would we conservative, since healthy migrants move to socio-economically disadvantaged areas? Why is the data used for the calculation of the deprivation index only available for 2001? Was the Population and Housing Census only conducted that year? The data section could be divided into two sections, one for “Socio-economic deprivation index” and another for “Causes of death”. Currently, the paragraph contains a lot of unrelated information. Although the authors later write that the deprivation index is dimensionless and arbitrary, one more sentence about it would be helpful for the reader’s understanding. Am I correct that each of the statistics are census-tract specific, but the index is normalized so that the mean across all tracts in all 7 cities is 0? Or is it normalized across cities themselves? Are the age-standardized mortality rates for adults only? And by 5-year age groups? Is there an age range or is it all adults? A possible useful additional analysis would be to re-estimate the analysis just for working age adults, since this age group might be facing the largest crisis-related stressors.

Results

Are the city-level all-cause ASMRs consistent with other estimates (if there are any)? Overall, the results discussion would benefit from more interpretation, rather than listing of results. Some sentences could be reworded to be more interpretive (something like “In Barcelona, residents of the most deprived census tracts were 2.5 times more likely to die of infectious and parasitic diseases”, etc). A possibility to restructure the paragraph (but it depends on the framing the authors are pursuing) would be to discuss results by city, rather than by cause of death. For example, what were notable patterns in Barcelona? Do other cities follow similar/different patterns? Perhaps there are several common patterns. This would also make the messaging more straightforward.

Discussion

Do the authors have any hypotheses on why the association develops for women in some cities, but not others? What do the 3 cities have in common? What about cities that don’t have the association? Is their overall mortality lower or higher than in cities with an association between deprivation and mortality? Is there an “ideal” city? Expand a little on lines 267-268. What are examples of long- and short-term effects?

Conclusion

The paper has interesting findings that could be highlighted in a stronger conclusion with clearer take-away messages.

Tables

Tables 1 & 2 should say somewhere that ASMR is per 100,000.

Reviewer 3 Report

Effect of the Financial crisis on socioeconomic inequalities in mortality in small areas in seven Spanish cities

Title seems odd to me. More specifically, the "socioeconomic inequalities in mortality." There are social inequalities which are associated with health inequalities (in this case mortality). In this case, socioeconomic inequalities is the result of the Financial crisis. So, I suggest editing the title and throughout the text to something along the lines of:

Effect of the financial crisis on inequalities in mortality in small areas in seven Spanish cities.

or

Effect of the financial crisis on health inequalities in mortality in small areas in seven Spanish cities.

In the introduction, the first line is odd, should be "the impact of economic crises on population health.

Methods:

How many census tracts were there per city? The units of analysis were Census Tracts, could the authors clarify how the Census tracts were incorporated in the analysis? Why not calculate the mortality rate for each of the cities? Was the nesting of the Census tracts within Cities taken into account by using multilevel modeling or a GEE? What was the Cronbach Alpha for the indicators of the socioeconomic deprivation index? I would like to know more about the internal consistency? Throughout the paper, the main exposure of interest should be described or listed before the health outcome. For example, on page 3, "To analyse the relationship between mortality and economic deprivation in three periods..." Should be "To analyse the relationship between economic deprivation and mortality..." the figures were difficult to read. They were very small.

Round 2

Reviewer 2 Report

All of my comments have been addressed. Thank you.